# Generally rare but occasionally severe weight gain after switching to an integrase inhibitor in virally suppressed AGE_hIV cohort participants

Sebastiaan O. Verboeket[1,2]*, Anders Boyd[3,4], Ferdinand W. Wit[1,2,3], Eveline Verheij[1,2], Maarten F. Schim van der Loeff[1,4], Neeltje Kootstra[5], Marc van der Valk[1], Peter Reiss[1,2,3], on behalf of the AGE_hIV Cohort Study Group¶

1 Department of Internal Medicine, Amsterdam Infection and Immunity Institute and Amsterdam Public Health Research Institute, Amsterdam UMC, University of Amsterdam, Amsterdam, The Netherlands, 2 Department of Global Health, Amsterdam UMC, University of Amsterdam and Amsterdam Institute for Global Health and Development, Amsterdam, The Netherlands, 3 HIV Monitoring Foundation, Amsterdam, The Netherlands, 4 Department of Infectious Diseases, Public Health Service of Amsterdam, Amsterdam, The Netherlands, 5 Department of Experimental Immunology, Amsterdam Infection and Immunity Institute, Amsterdam UMC, University of Amsterdam, Amsterdam, The Netherlands

¶ Complete membership of the author group can be found in the Acknowledgments.
* s.o.verboeket@amsterdamumc.nl

## Abstract

### Objectives

Recent studies have reported disproportionate weight gain associated with integrase strand transfer inhibitor (INSTI) initiation in antiretroviral therapy(ART)-naive people with HIV (PWH), particularly among black women. We investigated if HIV-positive AGE_hIV participants with suppressed viremia switching to INSTI-containing ART experienced more weight gain compared to HIV-positive virally-suppressed non-switching and HIV-negative controls.

### Methods

In the AGE_hIV cohort, standardized weight measurements were performed biennially. Participants switching to INSTI-containing ART were 1:2:2 propensity score-matched with controls by age, gender, ethnicity and body mass index. Mean weight changes and proportions experiencing >5% or >10% weight gain were compared between study-groups using linear mixed-effects models and logistic regression, respectively.

### Results

121 INSTI-switching participants and 242 participants from each of the control groups were selected. Across groups, median age was 53–55 years, 83–91% were male and 88–93% white. Mean weight change after switch among INSTI-switching participants was +0.14 kg/year (95%CI -0.25, +0.54) and similar among HIV-positive [+0.13 kg/year (95%CI +0.07, +0.33; P = .9)] and HIV-negative [+0.18 kg/year (95%CI 0.00, +0.37; P = .9)] controls.

**Data Availability Statement:** Data sharing has been restricted by the Ethical Review Board of the Amsterdam UMC (1) because the data underlying

this study contains very sensitive and potentially identifying information. Requests for data sharing can be made on a case-by-case basis following submission of a concept sheet as per instructions on the project website (https://agehiv.nl/en/science/). Once submitted the proposed research/analysis will undergo review for evaluation of the scientific value, relevance to the study, design and feasibility, statistical power and overlap with existing projects. If the proposed analysis is for verification/replication, data will then be made available. If the proposed research is for novel science, upon completion of the review, feedback will be provided to the proposer(s). In some circumstances, a revision of the concept may be requested. If the concept is approved for implementation, a writing group will be established consisting of the proposers (up to 3 persons that were centrally involved in the development of the concept) and members of the AGE_hIV Cohort Study group (or other appointed cohort representatives). All persons involved in the process of reviewing these research concepts are bound by confidentiality. (1) Medisch Ethische Toetsingscommissie Academisch Medisch Centrum, University of Amsterdam, Room TK0-270, Meibergdreef 9, 1105 AZ Amsterdam The Netherlands.

**Funding:** This work was supported by The Netherlands Organization for Health Research and Development (ZonMW, https://www.zonmw.nl/en/, grant number 300020007) and AIDS Fonds (https://aidsfonds.nl/, grant number 2009063). Additional unrestricted scientific grants were received from Gilead Sciences (https://www.gilead.com/); ViiV Healthcare (https://viivhealthcare.com/en-us/); Janssen Pharmaceuticals N.V. (https://www.janssen.com/); and Merck&Co (https://www.merck.com/). The funders had no role in study design, data collection and analysis, decision to publish, or preparation of the manuscript. All grands were received by PR.

**Competing interests:** PR through his institution has received independent scientific grant support from Gilead Sciences, Janssen Pharmaceuticals Inc, Merck & Co and ViiV Healthcare, and has served on scientific advisory boards for Gilead Sciences, ViiV Healthcare, Merck & Co, Teva pharmaceutical industries, for which honoraria were all paid to his institution. FWW has served on scientific advisory boards for ViiV and Gilead sciences. MFSVDL has received independent scientific grant support from Sanofi Pasteur, MSD Janssen Infectious Diseases and Vaccines and Merck, he has served on the advisory board of GSK and has received nonfinancial support from

Weight gain >5% occurred in 28 (23.1%) INSTI-switching, 38 HIV-positive (15.7%, P = .085) and 32 HIV-negative controls (13.2%, P = .018). Weight gain >10% was rare.

## Conclusions

Switching to INSTI-containing ART in our cohort of predominantly white men on long-term ART was not associated with greater mean weight gain, but >5% weight gain was more common than in controls. These results suggest that not all, but only certain, PWH may be particularly prone to gain a clinically significant amount of weight as a result of switching to INSTI.

## Introduction

International guidelines currently position integrase strand transfer inhibitors (INSTIs) as preferred agents for people with HIV (PWH) initiating combined antiretroviral therapy (cART), and ART-experienced PWH are frequently being switched to INSTIs from protease inhibitor (PI) or non-nucleoside reverse transcriptase inhibitor (NNRTI) based regimens [1, 2]. Most INSTIs are considered to have favorable pharmacological and toxicity profiles. However, multiple post-marketing studies have reported greater-than-expected weight gain among both cART-naive-[3–10] and treatment-experienced [11–16] PWH initiating an INSTI. Whether this will result in a similarly increased risk of obesity-related complications, like are found in the general population [17, 18] remains currently unclear.

The most striking data thus far were reported by the ADVANCE study [8–10], a randomized-controlled trial comparing the use of three cART regimens in ART-naive black South-African PWH: dolutegravir/tenofovir alafenamide(TAF)/emtricitabine, dolutegravir/tenofovir disoproxil/emtricitabine, and efavirenz/tenofovir disoproxil/emtricitabine. Use of the INSTI dolutegravir was significantly associated with greater weight gain, the effect of which was strongest in women and more prominent with concomitant use of TAF. Other studies reporting significantly greater mean weight gain in PWH initiating an INSTI generally describe more modest degrees of weight gain and INSTI-related weight gain to be particularly observed in specific sub-groups (i.e. women and people of black African descent) [3–7, 11–16]. A recent meta-analysis of randomized controlled trials conducted in ART-naive PWH reported that within the class of INSTIs, there was greater weight gain with the newer generation INSTIs dolutegravir and bictegravir compared to two previously licensed agents of this drug class, raltegravir and elvitegravir [6]. Some other studies, however, did not observe any above-normal weight gain among PWH initiating INSTIs [19–22].

The majority of studies thus far have examined weight gain in ART-naive PWH, [3–10, 21] for whom any INSTI-specific effect on weight is difficult to disentangle from the weight gain coinciding with initial suppression of HIV, as part of a 'return-to-health' phenomenon. Fewer studies have been published assessing PWH with suppressed viremia *switching* to an INSTI-containing regimen, in whom such a phenomenon is absent [11, 15, 22]. Only one of these studies included standardized weight measurements, with 62% and 25% of participants being overweight and obese prior to switch, respectively [15]. In order to expand our knowledge in persons switching to INSTI whilst already suppressed, we took advantage of data obtained in the AGE_hIV cohort study population, with 38% and 8%, respectively, of participants switching to INSTI being overweight and obese prior to switch. Furthermore, the AGE_hIV cohort only includes middle-aged and older PWH, in whom age-associated comorbidities were found to

Stichting Pathologie Onderzoek en Ontwikkeling. MVDV through his institution has received independent scientific grant support and consultancy fees from Abbvie, Gilead Sciences, Johnson & Johnson, MSD and ViiV Healthcare, for which honoraria were all paid to his institution. SOV, AB, EV, NAK and have no potential competing interests. The current work was supported by additional unrestricted scientific grants from Gilead Sciences; ViiV Healthcare; Janssen Pharmaceuticals N.V.; and Merck&Co. This does not alter our adherence to PLOS ONE policies on sharing data and materials.

be more prevalent among PWH compared to people without HIV of similar age [23]. INSTI-associated weight gain could therefore be particularly relevant for this older sub-population of PWH, as it could further increase risks of developing comorbidities, such as cardiovascular diseases or malignancies. Our aim was to determine if virally-suppressed HIV-positive participants switching to INSTI-containing cART experienced more weight gain compared to two propensity score matched control groups: (1) PWH with suppressed viremia who did not alter their cART regimen, and (2) HIV-negative participants.

## Materials and methods

### Study participants and data collection

The $AGE_hIV$ Cohort Study is an ongoing prospective cohort study evaluating the occurrence of age-related comorbidities in 598 HIV-1-positive and 550 HIV-negative participants. HIV-positive participants were recruited from the HIV outpatient clinic of the Amsterdam University Medical Centers, location Academic Medical Center. HIV-negative participants were recruited from the sexual health clinic at the Amsterdam Public Health Service and from the Amsterdam Cohort Studies on HIV/AIDS [24]. Participants were enrolled in 2010–2012 and included if at least 45 years of age. At each biennial study visit participants receive a standardized screening for age-related comorbidities and HIV-negative participants a 4th generation HIV-antibody test. Details about the study protocol have previously been published [23]. Detailed information on recent and historical HIV characteristics prior to, and during, study follow-up were obtained from the Dutch HIV Monitoring Foundation registry. This includes prospectively collected detailed data on prior and current use of ART, as well as reasons for regimen alterations [25]. Written informed consent was obtained from all participants and the study was approved by the ethical review board of the Amsterdam UMC and registered at ClinicalTrials.gov (identifier NCT01466582).

At each study visit, body weight was measured using electronic scales (Seca® type 877, Seca, Germany), which were calibrated annually. Participants were explicitly instructed to undress to underwear and socks and remove any heavy jewelry.

### Study group selection through propensity score matching

We identified all participants who switched to an INSTI-containing regimen during follow-up and had never used an INSTI before. Participants were subsequently included in the index group if (1) they had ≥1 weight measurement prior to and ≥1 weight measurement after switch, and (2) had an undetectable (<40 copies/mL) HIV-1 viral load ≥1 year prior to switch (allowing for isolated 'blips' up to 200 copies/mL). For comparison, we selected two separate control groups. Each index participant was matched to two HIV-positive non-switching and two HIV-negative control participants. Eligible HIV-positive non-switching controls were those who continued a PI- and/or NNRTI-based ART regimen (permitting changes in antiretroviral agents within classes and NRTI backbone). Potential HIV-negative controls were participants who remained HIV-negative during study follow-up. Controls with ≤1 weight measurement were excluded. (see S1 Fig).

As participants switched to INSTI-based cART at various time-points during follow-up, the goal was to identify a time-point at which subjects from the control group most closely resembled the index participant at the moment they had switched to INSTI. To accomplish this, we used a time-dependent propensity score [26] derived from the time-fixed covariates "ethnicity" (based on region of origin), gender, and the time-varying covariates age and body mass index (BMI). (see S2 Fig) These variables were selected a priori based on their known association with weight or risk of weight gain. A risk set was constructed in which the hazards of switching

to INSTI were modelled for all participants using a Cox proportional hazards model with the matching criteria as independent variables. Predicted hazards were estimated at the visit prior to switch for participants switching to INSTI and at each study visit during follow-up for control participants. Matched pairs were chosen by the smallest total distance in predicted hazards within matched sets. Controls were allowed to be matched with only one index-participant, and a match with an HIV-positive non-switching control was only allowed to occur if they had an HIV-RNA <40 copies/mL for >1 year while on cART.

Risk sets were constructed on information available at study visits. The actual date of switch occurred in-between study visits for the majority of participants in the group switching to INSTI. To select a date for matching controls, the number of days between date of study visit prior to switch and date of switch was first calculated in index participants. This offset was added to the date of the matched study visit in the control participants and was used as the hypothetical date of switch in these participants.

### Statistical analysis

Baseline was defined as the date of switch to INSTI-containing cART in the index participants and the date of hypothetical switch for control participants. We defined two follow-up periods: (1) *pre-baseline*, from the date of enrolment into the AGE$_h$IV cohort to baseline; and (2) *post-baseline*, from baseline until the last available AGE$_h$IV weight measurement, INSTI-discontinuation (for index participants), loss of HIV-RNA suppression (>40 copies/mL; excluding isolated blips <200 copies/mL), or death, whichever occurred first. The pre-baseline follow-up of HIV-positive participants started at the visit where a weight measurement was done and the HIV-RNA was suppressed, and all later visits also had HIV-RNA <40 copies/mL (excluding isolated blips up to 200 copies/mL).

We first used absolute body weight as an outcome, modelling mean yearly changes during follow-up by mixed-effects linear regression, in which between-participant variability at baseline and over time was accounted for by including a random intercept and slope, respectively. Mean changes in weight (i.e. interaction with time) between study-groups and pre- and post-baseline were directly calculated via a three-way interaction term. The differences in weight change slopes between study groups were statistically tested with a joint test using the 'contrast' command in Stata.

Subsequently, we used more prominent and potentially clinically-relevant weight gains as outcomes. These were defined as >5% (thus including those with >10%) or only >10% weight gain at the first weight measurement after baseline, using the last weight measured prior to baseline as comparison. As weight was measured biennially in all participants, choosing these measurements ensured comparable time-intervals during which weight gains could have occurred between study groups. The probability of this outcome was modeled using logistic regression and differences between study groups were tested using a Wald $X^2$ test.

Finally, we compared demographic and ART regimen-specific characteristics between groups of participants switching to INSTI experiencing three discrete categories of weight gain; those with 5–10% or >10% weight gain were compared to those with ≤5% weight gain using Fisher's exact and Wilcoxon rank-sum tests as appropriate. All statistical analyses were performed using Stata software (v12.0, College Station, TX, USA).

## Results

### Study group characteristics

From the 598 HIV-positive AGE$_h$IV participants, 212 had ever used an INSTI-containing regimen before their last available AGE$_h$IV cohort weight measurement. Of them, 121 fulfilled

criteria of an index participant switching to INSTI. Of these 121 participants, 64 (53%) switched to dolutegravir, 41 (34%) to elvitegravir and 16 (13%) to raltegravir during study follow-up. At switch, 60 (50%) participants also changed the nucleos(t)ide reverse transcriptase inhibitor (NRTI) backbone of their regimen (see S1 Table). Of note, no participants in the AGE$_h$IV cohort switched to a regimen including both dolutegravir and tenofovir alafenamide (TAF). The reasons for switching to INSTI differed, with regimen simplification being the most common (n = 41, 34%; see S2 Table).

From the eligible HIV-positive (N = 271) and HIV-negative (N = 488) controls, 242 HIV-positive non-switching participants and 242 HIV-negative participants were matched with index participants. Propensity score matching resulted in three comparable study groups with respect to age, body mass index (BMI), gender and ethnicity at the study visit prior to baseline (Table 1). The only significant difference was found in median age of HIV-negative controls compared to index-group participants (53 vs. 55 years respectively, $P$ = .02). HIV-specific characteristics, such as current and nadir CD4 count and time between diagnosis and ART

**Table 1. Characteristics of INSTI-switching, HIV-positive non-switching and HIV-negative selected study groups at baseline.**

| | INSTI-switching | HIV-positive non-switching | HIV-negative | $P$ INSTI switching vs. HIV-positive non-switchers | $P$ INSTI switching vs. HIV-negative |
|---|---|---|---|---|---|
| N | 121 | 242 | 242 | | |
| Time before baseline (yr) | 4.2 (3.6, 5.2) | 2.7 (1.2, 4.3) | 2.9 (1.5, 4.7) | <.001 | <.001 |
| Time after baseline (yr) | 1.9 (0.9, 2.8) | 2.2 (1.1, 4.0) | 3.0 (1.3, 4.6) | .007 | <.001 |
| Age (yr) | 55 (51, 61) | 54 (51, 61) | 53 (50, 59) | .7 | .02 |
| Male gender | 106 (88%) | 221 (91%) | 200 (83%) | .3 | .2 |
| MSM | 92 (79%) | 182 (79%) | 170 (71%) | .9 | .1 |
| Ethnic descent | | | | | |
| White[1] | 107 (88%) | 217 (90%) | 224 (93%) | .9 | .4 |
| African | 13 (11%) | 23 (10%) | 17 (7%) | | |
| Asian | 1 (1%) | 2 (1%) | 1 (0%) | | |
| BMI (kg/m$^2$) | 24.3 (22.4, 26.1) | 24.0 (22.2, 27.2) | 23.9 (22.5, 26.2) | .8 | .8 |
| BMI (kg/m$^2$) categories | | | | | |
| Underweight: <18.5 | 0 (0%) | 9 (4%) | 1 (0%) | .2 | .9 |
| Normal: 18.5–<25 | 75 (62%) | 137 (57%) | 152 (63%) | | |
| Overweight: 25-<30 | 36 (30%) | 74 (31%) | 70 (29%) | | |
| Obese: ≥30 | 10 (8%) | 22 (9%) | 19 (8%) | | |
| Smoking status | | | | | |
| Never | 38 (32%) | 64 (28%) | 95 (40%) | .07 | .4 |
| Former | 51 (43%) | 82 (36%) | 89 (37%) | | |
| Current | 29 (25%) | 85 (37%) | 54 (23%) | | |
| Latest CD4 count (cells/mm$^3$) | 640 (500, 790) | 630 (500, 840) | 870 (640, 1060) | .9 | <.001 |
| CD4 nadir (cells/mm$^3$) | 190 (75, 270) | 175 (90, 250) | . . . | .7 | . . . |
| Time since HIV diagnosis (yr) | 14 (8, 19) | 14 (10, 19) | . . . | .4 | . . . |
| Time since ART initiation (yr) | 12 (7, 17) | 13 (8, 16) | . . . | .9 | . . . |

Values are median (interquartile range) or n (%).

[1]Including Hispanic ethnicity P values were calculated using the Wilcoxon rank-sum test for continuous variables and the $X^2$ test for categorical variables.

Abbreviations: BMI, body mass index; MSM, men who have sex with men; ART, antiretroviral therapy; INSTI, integrase strand transfer inhibitor.

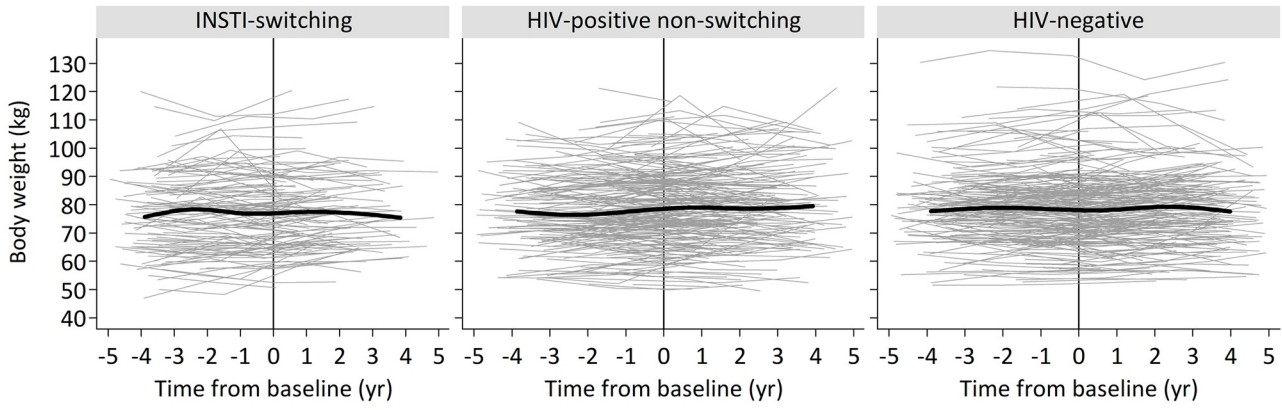

**Fig 1. Median and per participant body weight trajectories of INSTI-switching, HIV-positive non-switching and HIV-negative selected participants.** Grey lines demonstrate individual body mass index trajectories before and after baseline (i.e. moment of switch in INSTI-switching participants or at assigned hypothetical moment of switch in controls). Black lines are median splines (6 knots each) within each study group.

initiation, were not significantly different between HIV-positive study groups. Median follow-up after baseline was 1.9 (IQR = 0.9, 2.8), 2.2 (IQR = 1.1, 4.0) and 3.0 (IQR = 1.3, 4.6) years for HIV-positive index participants, HIV-positive controls and HIV-negative controls, respectively. Among the HIV-positive controls, 65 (27%) continued using a PI-, 168 (69%) an NNRTI-, and 9 (4%) a PI- plus NNRTI-containing regimen during follow-up.

## Mean weight changes before and after baseline

Fig 1 depicts both the body weight trajectories of each participant and the median body weight trajectories of the study groups, which were overall comparable. In the index group (Table 2), yearly changes in predicted mean weight were not significantly different from zero during both the pre- and post-baseline periods (0.15 kg/year, P = .2 and 0.14 kg/year, P = .5 respectively), nor statistically different between these follow-up periods (P = .9). Similarly, no significant differences in yearly changes of predicted mean weight were found between pre- versus post-baseline periods in the HIV-positive non-switching controls (0.08 vs. 0.13 kg/year respectively, P = .7). In the HIV-negative controls, mean yearly increase in weight was borderline significantly lower before baseline compared to after baseline (-0.06 vs. 0.18 kg/year respectively, P = .05). During the post-baseline follow-up period, the yearly changes in predicted mean weight were not significantly different when comparing the index group to either of the control groups (HIV-positive non-switching controls, P = .9; and HIV-negative controls, P = .9).

## Probability of experiencing >5% or >10% weight gain

Fig 2 illustrates the distribution of proportional weight changes. In this analysis, median time between baseline and post-switch weight measurement was 0.9 (IQR 0.4–1.5) years in the index group, 0.9 (IQR 0.4–1.6) years for HIV-positive controls (P = .7 vs. the index group) and 0.8 (IQR 0.4–1.5) years for HIV-negative controls (P = 0.2 vs. the index group). The probability of a >5% increase in weight after baseline was greater in participants switching to INSTI (n = 28, 23.1%) than in both HIV-negative (n = 32, 13.2%, P = .018) and in HIV-positive controls (n = 38, 15.7%, P = .085), with the latter comparison not reaching statistical significance. The probability of a >10% increase in weight was 5.0% (n = 6) for index participants, 3.7%, (n = 9, P = .6) for HIV-positive controls, and 2.5% (n = 5, P = .1) for HIV-negative controls.

**Table 2. Comparison of mean yearly changes in bodyweight by study group, before and after baseline.**

|  |  | Mean yearly change in body weight during follow-up periods | | | | Within group comparison, after vs. before baseline | |
|  |  | *Before* baseline | | *After* baseline | | | |
|  | N | kg/year | 95% CI | kg/year | 95% CI | Δ kg/year | 95% CI |
|---|---|---|---|---|---|---|---|
| INSTI-switching | 121 | 0.15 | -0.08, 0.39 | 0.14 | -0.25, 0.54 | -0.01 | -0.45, 0.43 |
| HIV-positive non-switching | 242 | 0.08 | -0.13, 0.29 | 0.13 | 0.07, 0.33 | 0.05 | -0.22, 0.32 |
| HIV-negative | 242 | -0.06 | -0.26, 0.13 | 0.18 | 0.00, 0.37 | 0.25 | 0.00, 0.50 |
| *Between group comparison* |  | *Before* baseline | | *After* baseline | |  |  |
|  |  | Δ kg/year | 95% CI | Δ kg/year | 95% CI |  |  |
| Switch to INSTI vs | 121 | Ref |  | Ref |  |  |  |
| HIV-positive non-switching | 242 | -0.07 | -0.39, 0.25 | -0.01 | -0.46, 0.43 |  |  |
| HIV-negative | 242 | -0.22 | -0.53, 0.09 | 0.04 | -0.40, 0.48 |  |  |

Reported results were calculated from a linear mixed-effects model with bodyweight as the dependent variable and a three-way interaction of study group x time x follow-up period (along the with the separate individual variables and two-way interactions between these variables) as independent variables. Abbreviations: CI, Confidence Interval; Δ, difference.

### Participants with ≤5%, 5–10% and >10% weight gain after INSTI switch

Three (50%) of the 6 index participants experiencing a >10% weight gain after baseline were black women, while there were only 2 (2%) black women among the 91 index participants with ≤5% weight gain after switch (P = .005, Table 3). In comparison, black women were not significantly more likely to have >10% versus ≤5% weight gain, respectively, in any of the control groups: 2 (7%) vs. 4 (2%) in HIV-positive non-switching controls (P = .3); and 1 (4%) vs. 7 (3%) in HIV-negative controls (P = .4). No specific INSTIs, or NRTIs were switched to more often among participants with >10% or 5–10% weight gain compared to those with ≤5% weight gain, nor were there any differences between having a PI- vs. NNRTI-based regimen prior to switch.

## Discussion

In our longitudinal analysis with standardized weight measurements, when judged by mean weight changes participants switching to INSTI-containing cART did not experience significantly greater weight gain compared to HIV-positive individuals who continued their non-INSTI-containing cART or HIV-negative individuals. Whereas other studies have reported significantly greater weight gains 0.9 to 2.4 years after switching to INSTI-containing cART compared to controls, the additional mean weight gain in these studies was limited to 0.05–2.2 kg [11], [12], [14–16], [22].

Of note, these previous studies focused on relatively small mean differences, rather than on the possible occurrence of more pronounced weight gain in a subset of participants. In our study, a 5% or greater weight gain after a median of 1 year was more likely among those who switched to INSTI-containing cART.

Hill et al have proposed to consider >5% weight gain a clinically relevant threshold when reporting INSTI- and more generally ART-associated increases in weight [27], mirroring 'clinically relevant' definitions for weight loss interventions by the United States Food and Drug Administration [28].

Our findings suggest INSTI-associated weight change to be heterogeneous, and provides support for the hypothesis that some individuals may be more prone to develop prominent

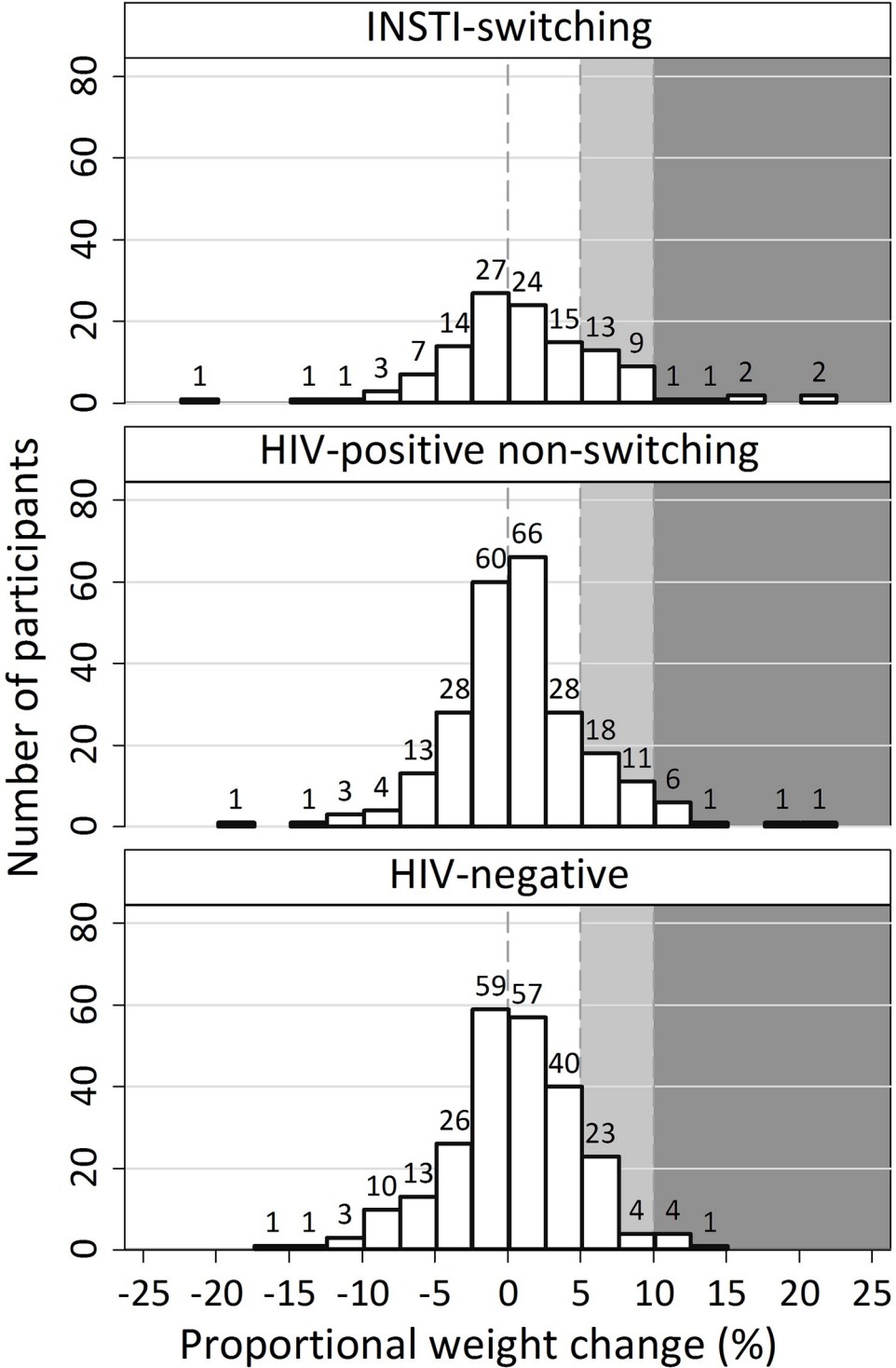

**Fig 2. Proportional weight change at first visit after baseline compared to last visit before baseline.** Change compares the first weight measurement after baseline with the last weight measured prior to baseline. Light and dark grey fields indicate participants with a proportional weight increase of >5% (light + dark grey) and >10% (dark grey). Numbers above bars indicate absolute number of participants per bin.

**Table 3. Characteristics of INSTI-switching participants with ≤5%, 5–10% and >10% weight gain after INSTI switch.**

| | ≤5% weight gain | 5–10% weight gain | >10% weight gain | P 5–10% vs. ≤5% weight gain | P >10% vs. ≤5% weight gain |
|---|---|---|---|---|---|
| N | 93 | 22 | 6 | | |
| Age at baseline (years) | 55 (50, 62) | 54 (52, 59) | 54 (53, 56) | .8 | 1.0 |
| Gender & Ethnicity | | | | | |
| Non-black male | 78 (85%) | 16 (73%) | 3 (50%) | .7 | .005 |
| Non-black female | 7 (8%) | 3 (14%) | 0 (0%) | | |
| Black male | 5 (5%) | 3 (14%) | 0 (0%) | | |
| Black female | 2 (2%) | 0 (0%) | 3 (50%) | | |
| BMI at baseline (kg/m$^2$) | 24.6 (22.8, 26.2) | 23.2 (21.1, 24.6) | 22.0 (19.8, 22.9) | .019 | .04 |
| INSTI initiated | | | | | |
| Dolutegravir | 46 (49%) | 14 (64%) | 4 (67%) | .5 | .9 |
| Elvitegravir | 34 (37%) | 5 (23%) | 2 (33%) | | |
| Raltegravir | 13 (14%) | 3 (14%) | 0 (0%) | | |
| Regimen prior to switch | | | | | |
| PI based | 39 (42%) | 13 (59%) | 3 (50%) | .5 | 1.0 |
| NNRTI based | 46 (49%) | 9 (41%) | 5 (50%) | | |
| Both NNRTI/PI | 5 (5%) | 0 (0%) | 0 (0%) | | |
| No NNRTI/PI | 3 (3%) | 0 (0%) | 0 (0%) | | |
| TDF after switch | 34 (37%) | 6 (27%) | 2 (33%) | .5 | 1.0 |
| TAF after switch | 19 (20%) | 4 (18%) | 2 (33%) | 1.0 | .6 |
| ABC after switch | 30 (32%) | 9 (41%) | 1 (17%) | .5 | .7 |

Comparisons were made using Wilcoxon rank-sum and Fisher's exact tests. Abbreviations: BMI, body mass index; INSTI, integrase-strand transfer inhibitor; PI, protease inhibitor; NNRTI, non-nucleoside reverse transcriptase inhibitor; TDF; tenofovir disoproxil; TAF, tenofovir alafenamide; ABC, abacavir.

INSTI-associated weight gain than others. The results of the ADVANCE trial and other studies which have shown that people of black African descent, particularly women, and women in general were at increased risk of INSTI-related weight gain and treatment-emergent obesity [8–10, 13, 15]. Interestingly, although black African women were clearly underrepresented in our study, they were overrepresented in the subgroup with more than 10% weight gain after switching to INSTI.

A mechanism underlying a specific susceptibility to prominent INSTI-associated weight gain is currently unknown, but it seems reasonable to speculate that pharmacogenetics may play a role. As stated in the European Medicines Agency assessment report for dolutegravir, dolutegravir in vitro was shown to inhibit binding of radiolabeled α-melanocyte-stimulating hormone (MSH) to the human recombinant melanocortin 4 receptor (MC4R) by 64% at a concentration equal to the clinical Cmax [29]. Whereas a more recent report largely confirmed this observation and in fact demonstrated a class-wide ability of INSTIs to bind MC4R, functional antagonistic effects in vitro were only observed at concentrations substantially greater than the therapeutic plasma concentrations of each drug [30].

The melanocortin system plays an important role in the regulation of food intake and body weight by the hypothalamus, and particular mutations and polymorphisms in the MC4R receptor gene have been demonstrated to increase the risk of obesity [31, 32]. Furthermore, individuals with a recessive single nucleotide polymorphism rs489693 near the MC4R gene, as well as those with polymorphisms in different genes expressed in other areas of the brain have been shown to increase susceptibility to extreme weight gain associated with the use of antipsychotics [33, 34]. Whether particular variants of the gene encoding MC4R, or other genes

involved in food intake and weight regulation, may be differentially present according to ethnicity and sex, and render individuals particularly susceptible to the potential effects of INSTI on proteins expressed by these genes at clinically relevant concentrations merits further investigation. A higher rate of genetic variation in the MC4R gene in people with African ancestry has previously been reported [35].

Importantly, the impact of a more pronounced weight gain on an individual's health can be expected to be greater, also dependent on prior weight and preexisting conditions, such as obesity and diabetes. Furthermore, the impact may well be influenced by other than biological factors. For example, African women have been shown to perceive obesity as less of a health threat than African men, and being obese can have more severe psychosocial effects for some ethnic groups than others [36, 37]. Thus, the extent to which individuals would be inclined to implement behavioral countermeasures such as diet and exercise, can be expected to be associated with socio-cultural perceptions of body weight. Further research is required to delineate the potential adverse cardiometabolic and psychological impact of INSTI-associated weight gain.

An important strength of this study was that body weight was measured in a standardized manner at pre-defined intervals both before and after switch, as opposed to studies which rely on weights captured using not regularly calibrated and varying scales, with inconsistent instructions to patients concerning disrobing. In addition, our study was not subject to bias which may occur when clinicians nowadays could be more inclined to measure weight (more frequently) in patients using INSTIs, given the increased interest in the subject of weight gain potentially associated with INSTIs. Finally, we were able to carefully match participants initiating INSTIs to both treated HIV-positive and HIV-negative control groups in a time-dependent fashion, further minimizing potential biases between comparison groups.

Our study nonetheless also has a number of limitations. First, the number of women, particularly black African women, and black African participants in general was limited, which precluded us from performing adequately powered analyses concerning the influence of gender and ethnic descent. Second, our sample size did not allow meaningful analyses into any differential effects of individual INSTIs including dolutegravir and bictegravir, or of the (concomitant) use of TAF as were observed in the ADVANCE trial and other studies [9, 10, 38]. Third, follow-up after switch to INSTI may have been insufficient for part of our study participants to allow INSTI-related weight gain to be observed. Finally, changes in weight can be caused by a host of other factors (e.g. smoking, smoking cessation, socio-economic status, mental health problems, etc.) and since many of these factors were either not collected or were fairly homogenous in our study population, they were not adjusted for in our analysis. Residual confounding from these factors could be present.

Generally speaking, these results are reassuring for the majority of PWH who consider switching their regimen to include an INSTI, in a country like the Netherlands with a largely white male HIV epidemic. In such a population the likelihood for someone to experience a prominent gain in weight appears to be a relatively rare event. This finding is relevant for older PWH, well-represented in our cohort, who are inclined to switch to an INSTI-containing regimen, for example to prevent potential drug-drug interactions with other co-medication taken for age-associated comorbidities. Nonetheless clinicians should remain vigilant in monitoring weight, particularly in women and black people. Assessing the mechanism by which some people are specifically prone to develop prominent weight gain on INSTI should be prioritized for further research.

## Supporting information

**S1 Fig. Flow-chart illustrating the selection of index and control participants from the AGE$_h$IV cohort.** [1]HIV viral load >40 copies/mL excluding 'blips' up to 200 copies/mL.

Abbreviations: INSTI, integrase strand transfer inhibitor; NNRTI, non-nucleoside reverse transcriptase inhibitor; cART, combination antiretroviral therapy; PI, protease inhibitor. (TIF)

**S2 Fig. Time-dependent propensity score matching of control participants to index-participants and determining their hypothetical moment of switch.** [1]HIV-positive non-switching or HIV-negative control. Propensity scores were calculated using a Cox proportional hazard model, including time-updated age and body mass index, and time-fixed gender and ethnicity. Abbreviations: INSTI, integrase strand transfer inhibitor. (TIF)

**S1 Table. Nucleoside reverse transcriptase inhibitor use before and after switch to INSTI.** (DOCX)

**S2 Table. Primary reasons for switching to INSTI.** (DOCX)

## Acknowledgments

**AGE$_h$IV Cohort Study Group**

**Scientific oversight and coordination:** P. Reiss (principal investigator), F.W.N.M. Wit, M. van der Valk, J. Schouten, K.W. Kooij, R.A. van Zoest, E. Verheij, S.O. Verboeket, B.C. Elsenga (Amsterdam University Medical Centers (Amsterdam UMC), University of Amsterdam, Department of Global Health and Amsterdam Institute for Global Health and Development (AIGHD)).

M. Prins (co-principal investigator), M.F. Schim van der Loeff, L. del Grande, V. Olthof, I. Agard (Public Health Service of Amsterdam, Department of Infectious Diseases).

**Datamanagement:** S. Zaheri, M.M.J. Hillebregt, Y.M.C. Ruijs, D.P. Benschop, A. el Berkaoui (HIV Monitoring Foundation).

**Central laboratory support:** N.A. Kootstra, A.M. Harskamp-Holwerda, I. Maurer, M.M. Mangas Ruiz, A.F. Girigorie, B. Boeser-Nunnink (Amsterdam UMC, Laboratory for Viral Immune Pathogenesis and Department of Experimental Immunology).

**Project management and administrative support:** W. Zikkenheiner, F.R. Janssen (AIGHD).

**Participating HIV physicians and nurses:** S.E. Geerlings, A. Goorhuis, J.W.R. Hovius, F.J.B. Nellen, T. van der Poll, J.M. Prins, P. Reiss, M. van der Valk, W.J. Wiersinga, M. van Vugt, G. de Bree, F.W.N.M. Wit; J. van Eden, A.M.H. van Hes, F.J.J. Pijnappel, A. Weijsenfeld, S. Smalhout, M. van Duinen, A. Hazenberg (Amsterdam UMC, Division of Infectious Diseases).

**Other collaborators:** P.G. Postema (Amsterdam UMC, Department of Cardiology); P.H.L. T. Bisschop, M.J.M. Serlie (Amsterdam UMC, Division of Endocrinology and Metabolism); P. Lips (Amsterdam UMC); E. Dekker (Amsterdam UMC, Department of Gastroenterology); N. van der Velde (Amsterdam UMC, Division of Geriatric Medicine); J.M.R. Willemsen, L. Vogt (Amsterdam UMC, Division of Nephrology); J. Schouten, P. Portegies, B.A. Schmand, G.J. Geurtsen (Amsterdam UMC, Department of Neurology); F.D. Verbraak, N. Demirkaya (Amsterdam UMC, Department of Ophthalmology); I. Visser (Amsterdam UMC, Department of Psychiatry); A. Schadé (Amsterdam UMC, Department of Psychiatry); P.T. Nieuwkerk, N. Langebeek (Amsterdam UMC, Department of Medical Psychology); R.P. van Steenwijk, E. Dijkers (Amsterdam UMC, Department of Pulmonary medicine); C.B.L.M. Majoie, M.W.A. Caan (Amsterdam UMC, Department of Radiology); H.W. van Lunsen, M.A.F. Nievaard

(Amsterdam UMC, Department of Gynaecology); B.J.H. van den Born, E.S.G. Stroes, (Amsterdam UMC, Division of Vascular Medicine); W.M.C. Mulder, S. van Oorspronk (HIV Vereniging Nederland).

**Disclaimer:** These data were presented in part as oral presentations during the 21st International Workshop on Co-morbidities and Adverse Drug Reactions in HIV and the 17th European AIDS conference in Basel, Switzerland, November 2019.

## Author Contributions

**Conceptualization:** Sebastiaan O. Verboeket, Anders Boyd, Ferdinand W. Wit, Peter Reiss.

**Data curation:** Sebastiaan O. Verboeket, Ferdinand W. Wit.

**Formal analysis:** Sebastiaan O. Verboeket, Anders Boyd.

**Funding acquisition:** Peter Reiss.

**Investigation:** Sebastiaan O. Verboeket.

**Methodology:** Sebastiaan O. Verboeket, Anders Boyd, Ferdinand W. Wit.

**Project administration:** Sebastiaan O. Verboeket.

**Supervision:** Anders Boyd, Peter Reiss.

**Validation:** Sebastiaan O. Verboeket.

**Visualization:** Sebastiaan O. Verboeket, Anders Boyd.

**Writing – original draft:** Sebastiaan O. Verboeket, Anders Boyd, Ferdinand W. Wit, Peter Reiss.

**Writing – review & editing:** Sebastiaan O. Verboeket, Anders Boyd, Ferdinand W. Wit, Eveline Verheij, Maarten F. Schim van der Loeff, Neeltje Kootstra, Marc van der Valk, Peter Reiss.

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
