## [Decision Letter · Decision Letter 0]

1 Dec 2020

PONE-D-20-29715

Rare but occasionally severe weight gain after switching to an integrase inhibitor in virally suppressed AGEhIV cohort participants

PLOS ONE

Dear Dr. Verboeket,

Thank you for submitting your manuscript to PLOS ONE. After careful consideration, we feel that it has merit but does not fully meet PLOS ONE’s publication criteria as it currently stands. Therefore, we invite you to submit a revised version of the manuscript that addresses the points raised during the review process.

We look forward to receiving your revised manuscript.

Kind regards,

Robert Güerri-Fernández, MD, PhD

Academic Editor

PLOS ONE

Journal Requirements:

'This work was supported by The Netherlands Organization for Health Research and Development (ZonMW, https://www.zonmw.nl/en/, grant number 300020007) and AIDS Fonds (https://aidsfonds.nl/, grant number 2009063). Additional unrestricted scientific grants were received from Gilead Sciences (https://www.gilead.com/); ViiV Healthcare (https://viivhealthcare.com/en-us/); Janssen Pharmaceuticals N.V. (https://www.janssen.com/); and Merck&Co (https://www.merck.com/). The funders had no role in study design, data collection and analysis, decision to publish, or preparation of the manuscript. All grands were received by PR.'

We note that you received funding from a commercial sources: Gilead Sciences, ViiV Healthcare, Janssen Pharmaceuticals N.V. and Merck&Co.

Within this Competing Interests Statement, please confirm that this does not alter your adherence to all PLOS ONE policies on sharing data and materials by including the following statement: "This does not alter our adherence to PLOS ONE policies on sharing data and materials.” (as detailed online in our guide for authors http://journals.plos.org/plosone/s/competing-interests).  If there are restrictions on sharing of data and/or materials, please state these.

Please note that we cannot proceed with consideration of your article until this information has been declared.

Additional Editor Comments:

After a careful review of this paper I agree with the reviewers that this is a very good work. I would suggest you to answer the reviews and I would be glad to reconsider it to publication.

Reviewers' comments:

Reviewer's Responses to Questions

**Comments to the Author**

1. Is the manuscript technically sound, and do the data support the conclusions?

Reviewer #1: Yes

Reviewer #2: Yes

2. Has the statistical analysis been performed appropriately and rigorously? 

Reviewer #1: Yes

Reviewer #2: Yes

3. Have the authors made all data underlying the findings in their manuscript fully available?

Reviewer #1: Yes

Reviewer #2: Yes

4. Is the manuscript presented in an intelligible fashion and written in standard English?

Reviewer #1: Yes

Reviewer #2: Yes

5. Review Comments to the Author

Reviewer #1: The article is concise and well written. Even though the statistical approach is simple, it is clear and goes to the point of the article. The topic is quite relevant and actually it gives important information on a controversial topic. The sample size is not so big but enough for what they want to assess, at least in this small study.

The results are striking and they give new information that could help clinicians on their general practice, or at least to think about it and rethink what it has been said before, and probably larger studies on this matter will be necessary to confirm the data.

It is a good start point for further studies.

Reviewer #2: First of all I would like to mention that the work done is good, very methodical and correct. In general, it seems to me that the objectives of the paper today cover a very interesting and tendentious topic.

Major comments:

I believe my most important comment is about the population evaluated in this study. I think that the authors explain and justify very well the fact that women and population with African ethnicity are underrepresented in their study, but they do not give relevance and forget to mention the role of weight gain in the population that does represent them, the AGEhIV cohort. I think that the role that age can have as an independent factor in weight gain is not considered and I think that this could have a potential exploitation given the anthropomorphic changes that the aging population experiences, HIV negative and HIV positive. Especially if the cohort that is being evaluated aims to assess comorbidities in the aging of the HIV population.

Minor comments:

- The abstract is excellent, informative and concise.

- The scientific background is well documented, with references of the most important studies up to date. Once again I suggest mentioning age and weight gain relation. I also would recommend including the hypotheses in the introduction.

- In the methods section, the setting and design of the study are well explained, as is the selection of participants and follow-up. The matching criteria are clear. I recommend explaining the potential confounders, and effect modifiers.

- The results are clear; I have not found errors in the tables.

- I really like the discussion. But I repeat for the third and last time, I think it would be interesting in this paper to give relevance to age since it would make this work different from other papers and studies already published.

6. PLOS authors have the option to publish the peer review history of their article (what does this mean?). If published, this will include your full peer review and any attached files.

Reviewer #1: No

Reviewer #2: **Yes: **Lorena de la Mora Cañizo

---

## [Author Response · Author response to Decision Letter 0]

2 Feb 2021

Editorial Comments: After a careful review of this paper I agree with the reviewers that this is a very good work. I would suggest you to answer the reviews and I would be glad to reconsider it to publication. 

Response: We thank the editor for the compliment and the opportunity to revise our manuscript.

Reviewer #1: The article is concise and well written. Even though the statistical approach is simple, it is clear and goes to the point of the article. The topic is quite relevant and actually it gives important information on a controversial topic. The sample size is not so big but enough for what they want to assess, at least in this small study.

The results are striking and they give new information that could help clinicians on their general practice, or at least to think about it and rethink what it has been said before, and probably larger studies on this matter will be necessary to confirm the data.

It is a good start point for further studies.

Response: We thank the reviewer for these positive and supportive comments. We agree further research is needed to study the exact nature of the relationship between INSTI initiation and changes in body weight, including which patient characteristics may put them at increased risk of experiencing clinically significant increases in weight.

Reviewer #2: First of all I would like to mention that the work done is good, very methodical and correct. In general, it seems to me that the objectives of the paper today cover a very interesting and tendentious topic.

Response: We thank the reviewer for these positive and supportive comments.

Major comments:

I believe my most important comment is about the population evaluated in this study. I think that the authors explain and justify very well the fact that women and population with African ethnicity are underrepresented in their study, but they do not give relevance and forget to mention the role of weight gain in the population that does represent them, the AGEhIV cohort. I think that the role that age can have as an independent factor in weight gain is not considered and I think that this could have a potential exploitation given the anthropomorphic changes that the aging population experiences, HIV negative and HIV positive. Especially if the cohort that is being evaluated aims to assess comorbidities in the aging of the HIV population.

Response: We thank the reviewer for suggesting to stress that the AGEhIV cohort is indeed more representative of a middle-aged / older population of PWH. The issue of weight gain is especially relevant for this older population of PWH, as INSTI-related increase in overweight may further contribute to an already increased risk of developing comorbidities. We have now put more emphasis on these arguments in the introduction (lines 87-91 [Furthermore, the … diseases or malignancies.]) and discussion (lines 321-323 [This finding … age-associated comorbidities.]). 

We strongly agree with the reviewer that advancing age plays an important role in bodyweight changes and other measures of body composition. We therefore selected age as one of the matching criteria in our analysis to minimize any confounding bias due to age. 

The reviewer suggests here and in following comments to also analyze the relationship between age and weight gain in our data. Although the reviewer raises an interesting point, it is important to stress that our main research question was to study the effect on weight after switching to INSTI independent of important confounders including age. Moreover, of note only Individuals over the age of 45 were included in the AGEhIV Cohort, resulting in the large majority of participants falling within a narrow age band. In fact, the interquartile range for age was between 50 and 60 years for participants included in the current analysis. This precludes us from comparing differences in weight gain following switch to INSTI across a wide spectrum of ages. 

If we look at mean weight gain during study follow-up in the current study (table 2, i.e. weight gain with advancing age), there was only a maximum of 100-200 grams per year weight gain, i.e. a rather limited weight change. These results are in accordance with other studies among both HIV-positive and HIV-negative populations, which show on average weight to increase with increasing age, but to plateau or even decrease from around 50 or 60 years of age, especially among men.(1-3) This shows the importance of adjusting - or in our case matching – for age as it can be an important potential confounder the influence of which on the association between INSTI and weight gain could be non-linear. 

Minor comments:

- The abstract is excellent, informative and concise.

Response: We thank the reviewer for this positive comment

- The scientific background is well documented, with references of the most important studies up to date. Once again I suggest mentioning age and weight gain relation. I also would recommend including the hypotheses in the introduction.

Response: We thank the reviewer for these positive comments. The relationship between being overweight and having a higher risk of developing comorbidities is now discussed in the introduction by the addition of lines 87-91 [Furthermore, the … diseases or malignancies.]. For further discussion regarding studying the relationship between age and weight, please refer to the previous comments. 

- In the methods section, the setting and design of the study are well explained, as is the selection of participants and follow-up. The matching criteria are clear. I recommend explaining the potential confounders, and effect modifiers.

Response: The potential confounders age, BMI, ethnicity and sex were accounted for using the time-updated propensity score matching approach. We therefore limited adjusting in the analysis to these confounders. These potential confounders were chosen a priori, which is now discussed in lines 128-129 [These variables … weight gain]. There is a large number of other potential confounders such as levels of physical activity, mental health issues, co-medication, smoking, cessation of smoking, alcohol use, dietary changes, socio-economic status, etc. We have now added the possibility for residual confounding by other unmeasured factors in the limitations section (lines 312-317 [Finally, changes ... be present.]).

Our pre-specified analysis plan only specified a three-way interaction between time, follow-up period, and HIV-status, which was used to obtain the parameter estimates for weight change over time within strata of period (pre- and post- baseline) and HIV-status (HIV-positive and HIV-negative). No other interactions (i.e. effect modification) were studied.

- The results are clear; I have not found errors in the tables.

Response: We thank the reviewer for this positive comment.

- I really like the discussion. But I repeat for the third and last time, I think it would be interesting in this paper to give relevance to age since it would make this work different from other papers and studies already published.

Response: We thank the reviewer for this positive comment. As discussed in response to the reviewer’s first comment, we have included a sentence about the relevance of age with respect to INSTI-related weight gain in the discussion (lines 321-323 [This finding … age-associated comorbidities.]).

References

1. Erlandson KM, Zhang L, Lake JE, Schrack J, Althoff K, Sharma A, et al. Changes in weight and weight distribution across the lifespan among HIV-infected and -uninfected men and women. Medicine (Baltimore). 2016;95(46):e5399.

2. Jacobsen BK, Melhus M, Kvaloy K, Siri SRA, Michalsen VL, Broderstad AR. A descriptive study of ten-year longitudinal changes in weight and waist circumference in the multi-ethnic rural Northern Norway. The SAMINOR Study, 2003-2014. PLoS One. 2020;15(2):e0229234.

3. Williamson DF. The 10-Year Incidence of Overweight and Major Weight Gain in US Adults. Archives of Internal Medicine. 1990;150(3):665.

---

## [Editor Report · Decision Letter 1]

22 Apr 2021

Generally rare but occasionally severe weight gain after switching to an integrase inhibitor in virally suppressed AGEhIV cohort participants

PONE-D-20-29715R1

Dear Dr. Verboeket,

We’re pleased to inform you that your manuscript has been judged scientifically suitable for publication and will be formally accepted for publication once it meets all outstanding technical requirements.

Kind regards,

Giordano Madeddu

Academic Editor

PLOS ONE

---

## [Editor Report · Acceptance letter]

26 Apr 2021

PONE-D-20-29715R1 

Generally rare but occasionally severe weight gain after switching to an integrase inhibitor in virally suppressed AGEhIV cohort participants 

Dear Dr. Verboeket:

I'm pleased to inform you that your manuscript has been deemed suitable for publication in PLOS ONE. Congratulations! Your manuscript is now with our production department. 

Kind regards, 

on behalf of

Dr. Giordano Madeddu 

Academic Editor

PLOS ONE